# SIDT2 Associates with Apolipoprotein A1 (ApoA1) and Facilitates ApoA1 Secretion in Hepatocytes

**DOI:** 10.3390/cells12192353

**Published:** 2023-09-26

**Authors:** Alicia Sampieri, Alexander Asanov, Kevin Manuel Méndez-Acevedo, Luis Vaca

**Affiliations:** 1Departamento de Biología Celular y del Desarrollo, Instituto de Fisiología Celular, Universidad Nacional Autónoma de México, México City 04510, Mexico; asampier@ifc.unam.mx; 2TIRF Labs, 106 Grendon Place, Cary, NC 27519, USA; alexander.asanov@gmail.com; 3University of Cambridge Metabolic Research Laboratories and NIHR Cambridge Biomedical Research Centre, Wellcome-MRC Institute of Metabolic Science, Addenbrooke’s Hospital, Cambridge CB2 0QQ, UK; kmma_13@hotmail.com

**Keywords:** SIDT2, ApoA1, cholesterol, CRAC motif

## Abstract

SIDT2 is a lysosomal protein involved in the degradation of nucleic acids and the transport of cholesterol between membranes. Previous studies identified two “cholesterol recognition/interaction amino acid consensus” (CRAC) motifs in SIDT1 and SIDT2 members. We have previously shown that the first CRAC motif (CRAC-1) is essential for protein translocation to the PM upon cholesterol depletion in the cell. In the present study, we show that SIDT2 and the apolipoprotein A1 (ApoA1) form a complex which requires the second CRAC-2 motif in SIDT2 to be established. The overexpression of SIDT2 and ApoA1 results in enhanced ApoA1 secretion by HepG2 cells. This is not observed when overexpressing the SIDT2 with the CRAC-2 domain mutated to render it unfunctional. All these results provide evidence of a novel role for SIDT2 as a protein forming a complex with ApoA1 and enhancing its secretion to the extracellular space.

## 1. Introduction

RNA interference (RNAi) in the nematode *C. elegans* is a systemic process, meaning that the injection of double-stranded RNA (dsRNA) into one tissue leads to the posttranscriptional silencing of that specific gene in other tissues [1]. A genome-wide screening searching for genes involved in systemic RNAi led to the identification of a systemic RNA interference-deficient (sid-1) gene [2]. A search in the gene databank identified two mammalian candidates for SID-1 homologues with very low homology to SID-1 [2]. The identification of mammalian systemic RNA interference-deficient genes is puzzling since mammals do not show systemic RNAi. Later, we showed that higher sequence homology was found between mammalian SIDT1 and SIDT2 genes and the previously uncharacterized *C. elegans* gene, ZK721.1/tag-130 [3]. We also showed that ZK721.1/tag-130 is essential for cholesterol uptake in the nematode [3]. Furthermore, we identified a “cholesterol recognition/interaction amino acid consensus” (CRAC) motif in ZK721.1/tag-130, which interacts directly with cholesterol [3]. Disrupting this CRAC motif with a single amino acid mutation prevents cholesterol association to the protein [3]. These results led us to hypothesize that mammalian SIDT1 and SIDT2 genes may be involved in cholesterol transport. In a follow-up study, we showed that both mammalian proteins indeed interact directly with cholesterol via two CRAC motifs, one present in the extracellular loop and the second motif present in one of the transmembrane domains [4]. Mammalian SIDT proteins are present in intracellular organelles, including lysosomes [4]. SIDT1 translocates to the plasma membrane upon the depletion of cholesterol in living cells [4].

SIDT proteins play a variety of physiological roles, including lipid hydrolytic activity (SIDT2 hydrolyzes ceramide into sphingosine and fatty acid) [5], the uptake of RNA and DNA and degradation by the lysosome [6], apoptosis and autophagy [7]. The lysosomal-autophagic pathway plays an important role in hepatocyte lipid metabolism, SIDT2 knockout mice show a blockade of autophagosome maturation and lipid accumulation in the liver [7].

Further evidence supporting a role for SIDT2 in cholesterol metabolism/transport comes from our recent genome-wide association study, where we identified a SIDT2 functional variant present in the Mexican population associated with high HDL-C levels and reduced risk of premature coronary artery disease (CAD) [8]. This variant was also associated with increased levels of ApoA1 in the serum of individuals expressing this SIDT2 variant [8].

The finding of increased serum levels for ApoA1 in individuals expressing this SIDT2 variant prompted us to investigate the possible functional relationship between SIDT2 and ApoA1.

In the present study, we identified a novel role for SIDT2, showing that this protein associates with apolipoprotein A1 (ApoA1) and participates in the modulation of high-density lipoprotein (HDL-C) cholesterol levels.

ApoA1 is the major component of HDL-C and participates in the production of high-density lipoprotein (HDL) particles [9]. HDL-C plays a critical role in the reverse cholesterol transport (RCT), a process that brings cholesterol from peripheric tissues into the liver [10]. Proteomic analysis HDL-C identified many components in these particles, including apolipoproteins, lipid transport proteins, acute-phase response proteins, and proteinase inhibitors, among others [9,11,12]. Interestingly, SIDT2 is not present in the HDL-C particles.

HDL-C plays a crucial role in cardiovascular disease; an inverse correlation between HDL-C levels and risk of cardiovascular disease is well established [13], although very high levels of HDL-C could be deleterious [14].

ApoA1 is produced and secreted by the intestine and liver [9]. ApoA1 collects cholesterol via a direct interaction with the ATP-binding cassette (ABC) transporter A1 (ABCA1) located at the plasma membrane of cells [15,16].

The exact mechanism involved in the secretion of ApoA1 by hepatocytes is poorly understood, but a fraction of secreted ApoA1 is in lipidated state while the remaining ApoA1 is secreted in a lipid-free form [17,18]. The mechanism by which lipid-free ApoA1 is secreted remains unknown.

In the present study, we show that SIDT2 and ApoA1 associate in intracellular compartments and move to the plasma membrane upon cholesterol depletion. The transmembrane CRAC domain from SIDT2 is essential for the association to ApoA1, since disrupting this motif prevents SIDT2-ApoA1 interactions.

The expression of SIDT2 in human hepatocytes increases the amount of ApoA1 secreted to the medium. The expression of SIDT2 with the transmembrane CRAC domain mutated (SIDT2-mut) reduces the secretion of ApoA1. Furthermore, RNA interference (RNAi) experiments directed to knockdown SIDT2 using a small interference RNA show a substantial reduction in endogenous ApoA1 secreted by human hepatocytes.

All these results provide evidence of a novel role for SIDT2 in the transport of ApoA1 and suggest a possible role for SIDT2 in HDL-C regulation.

## 2. Materials and Methods

### 2.1. Cell Culture

All reagents used in this study were of analytical grade and purchased from Sigma (Saint Louis, MO, USA) unless otherwise stated. Human embryonic kidney (HEK293T) and HepG2 cells were purchased from the American Type Culture Collection (ATCC, Manassas, VA, USA). Cells were grown on 35 mm culture dishes using DMEM (Invitrogen, Carlsbad, CA, USA) supplemented 10% of heat-inactivated Fetal Bovine Serum (Wisent, premium quality, Saint-Jean-Baptiste, QC, Canada), Penicillin-Streptomycin (Life technologies, Carlsbad, CA, USA) and glutamine (Life technologies) in an incubator with humidity control at 37 °C and 5% of CO_2_.

Human *SIDT2* was cloned from human cDNA CGI-40 (AF151999.1) obtained from the Riken Consortium (Tokyo, Japan). The product was cloned in pEGFP-N1 (Clontech, Mountain View, CA, USA).

The plasmids containing SIDT2-GFP [4] and ApoA1-OFPSpark (Sino Biological, Wayne, PA) were transfected to HEK293T or HepG2 cells grown on 35 mm Petri dishes with a glass bottom (MatTek, Ashland, MA, USA). For transfection, we used a mixture of 1 µg of DNA and 6 µL CaCl_2_ reaching 60 µL of final volume with distilled H_2_O. The mixture was added by dropping to HeBS buffer (50 mM HEPES, 280 mM NaCl, 1.5 mM Na_2_HPO_4_, pH 7.05) and the final mixture was incubated for 30 min prior to replacing with DMEM. The cells were incubated overnight with the transfection mixture, which was then replaced with fresh medium to perform the assays 24 h later.

### 2.2. Constructs Used in This Study

SIDT2 cDNA was obtained from the Riken Consortium (Tokyo, Japan) and fused to the green fluorescent protein cDNA (GFP) as previously described [4]. The SIDT2-mut where tyrosine 650 was replaced by alanine was produced using the QuikChange^®^ Site-Directed Mutagenesis Kit (Stratagene, Santa Clara, CA, USA) as previously described [4]. ApoA1-OFPSpark was purchased from (Sino Biological, Wayne, PA, USA).

### 2.3. Western Blot Analysis of SIDT2 and ApoA1

For the evaluation of SIDT2 and ApoA1 on Western blot, we grew confluent monolayers of HEK293T cells on 10 petri dishes with diameter of 100 mm for each condition. HEK293T cells were transfected with 1 µg of the plasmids carrying SIDT2-GFP and ApoA1. Identification of SIDT2-GFP was conducted using an anti-GFP antibody (A-11122) from ThermoFisher Scientific (Waltham, MA, USA). For ApoA1, the anti-ApoA1 monoclonal antibody (513) was utilized ThermoFisher Scientific (Waltham, MA, USA). Protein biotinylation was conducted using the Biotin Protein Labeling Kit (Sigma-Aldrich, Saint Louis, MO, USA) to identify extracellularly exposed SIDT2-GFP and ApoA1 proteins. Biotin-labeled proteins were purified according to the instructions provided by the manufacturer and prepared for SDS-PAGE electrophoresis analysis. A total of 100 µg of total membrane extracts from cells expressing SIDT2-GFP and ApoA1 proteins were resolved using SDS-PAGE electrophoresis and transferred to nitrocellulose paper, as previously reported [4]. After 30 min of incubation at 37 °C, the nitrocellulose sheets were washed 3 times for 15 min each time with buffer and incubated for 30 min with luciferase-labeled mouse anti-rabbit antibody diluted 1:1000 (Sigma-Aldrich, Saint Louis, MO, USA). After a second washing period, bound antibody was visualized with Supersignal chemiluminescence detection kit from (ThermoFisher Scientific, Waltham, MA, USA). For endogenous SIDT2 detection in HepG2 cells, the anti-SIDT2 polyclonal antibody (PA5-69064) was used (ThermoFisher Scientific, Waltham, MA, USA). Full-length blots are included in Appendix A.

### 2.4. Confocal Microscopy and Förster Resonance Energy Transfer (FRET) Studies between SIDT2 and ApoA1

The plasmids containing SIDT2-GFP and ApoA1-OFPSpark were transfected to HEK293T cells grown on 35 mm Petri dishes with a glass bottom (MatTek, Ashland, MA, USA). Confocal microscopy images were collected with an Olympus Fv10i confocal microscope equipped with a UPLSAPO 60X 1.35 NA oil immersion objective using the following exciting wavelengths: SIDT2-GFP 488 nm, ApoA1-OFPSpark 570 nm. Emission was collected at 525 nm and 590 nm, respectively. For plasma membrane colocalization studies we used the plasma membrane dye FM4-64 ThermoFisher Scientific (Waltham, MA, USA). The excitation wavelength for FM4-64 was 488 nm, and emission was collected at 700 nm.

For FRET studies between SIDT2-GFP and ApoA1-OFPSpark, we utilized the acceptor photobleaching method. Briefly, a square ROI of about 20% of visible cell cytoplasm was selected and bleached using the Tornado function (40%, 570 nm; 1.5 s) in an Olympus FV-1000 Confocal Microscope equipped with a 60X 1.45 NA oil immersion objective. Three consecutive images were taken before and after acceptor photobleaching of OFPSpark and basal decay of fluorophores was assessed from the not-photobleached area of the same cell. FRET Efficiency is given by the equation: FRETeff = (Dpost/Dpre)/Dpost (D = donor intensity). FRET efficiency was calculated with Olympus ASW 2.1 Software.

For line plot analysis, we used the plot profile plugin from ImageJ. Briefly, time courses of fluorescence from HEK293T cells expressing SIDT2-GFP and ApoA1-OFPSpark were acquired with a time interval of 10 s/frame to minimize photobleaching. For the first 5 min, no treatment was applied to obtain a basal fluorescence distribution. After this control period, the cells were treated with 5 mM of methyl-b-cyclodextrin (MbCD) and image acquisition continued for several minutes until changes in protein cellular localization were evident (typically around 20–30 min).

### 2.5. RNA Interference of SIDT2 in HepG2 Cells

The cDNA from human SIDT2 (NM_001040455) was uploaded in the GenScript siRNA target finder (GenScript, Piscataway, NJ, USA). From the candidates provided, we selected 1 with the lowest off-target score (GACCGAAGACCAAGCCTGC). The candidate was synthesized as small interference RNAs (siRNA) (GenScript, Piscataway, NJ, USA). As a control, a scrambled siRNA was synthesized also (CCAGGAACACAGAGCCGTC). A Petri dish with 50% confluent HepG2 monolayer was transfected with the siRNA designed for SIDT2 (siSIDT2) or the scrambled siRNA (siScrambled). Cells were transfected using a previously published protocol [19]. Briefly, in a first tube 600 pmoles of the siRNAs was diluted into 500 μL Opti-MEM; in a second tube, 15 μL Lipofectamine was diluted into 500 μL Opti-MEM. The tubes were incubated at room temperature for 5 min. First and second tubes were mixed and incubated for 25 min. Culture media from HepG2 monolayers were removed, and the mixture solutions containing the siRNAs were added, leading to a siRNA final concentration of 300 nM. The cells were incubated at 37 °C in an incubator with 5% CO_2_ overnight. The siRNA transfection procedure was repeated 24 h later. Twenty-four hours after the second transfection, cells were rinsed twice and prepared for Western blot analysis and the ELISA detection of secreted ApoA1 (see below).

### 2.6. Measuring the ApoA1 Secreted by HepG2 Cell Cultures

HepG2 is a cell line exhibiting epithelial-like morphology that was isolated from a human hepatocellular carcinoma. HepG2 was purchased from the American Type Culture Collection (ATCC, Manassas, VA, USA). For quantification of endogenous ApoA1 secretion, we used the ApoA1 ELISA kit (ThermoFisher Scientific, Waltham, MA, USA) following the manufacturer’s instructions provided with the manual (Catalog Number EHAPOA1). Briefly, media obtained from 100 mm Petri dishes containing >80% confluent HepG2 monolayers were collected overnight (14 h). Cells were maintained in serum-reduced Opti-MEM medium (ThermoFisher Scientific, MA, USA). One Petri dish was used as a control (untransfected). A second Petri dish was transfected with scrambled siRNA and the third dish with an siRNA developed for human SIDT2 (siSIDT2). For the transfection protocol please refer to the RNA interference procedure above. A total of 1 mL of HepG2 medium for each condition collected overnight was diluted 1X with Assay Diluent provided with the kit. A total of 100 μL of HepG2 diluted medium was incubated on each well from the ELISA kit (5 replicates for each condition). The ELISA plate was covered and incubated for 3 h at room temperature with gentle shaking. After the incubation period, the solution was removed, and wells were washed 4 times with 1X Wash Buffer (300 μL) using a multi-channel Pipette. After the last wash, all solution was removed by decanting. A total of 100 μL of biotin conjugate was added to each well and incubated for 1 h at room temperature with gentle shaking. After the incubation period, the solution was removed and each well washed 3 times as described above. A total of 100 μL of Streptavidin-HRP solution provided with the ELISA kit (Streptavidin-HRP was diluted 1000-fold with 1X Assay Diluent) was added to each well and incubated for 45 min at room temperature with gentle shaking. After the incubation period, the solution was removed and each well washed 3 times as described above. A total of 100 μL of TMB Substrate (provided with the kit) was added to each well, and the plate was incubated for 30 min at room temperature in the dark with gentle shaking. After this period, 50 μL of Stop Solution was added to each well. The absorbance at 450 nm was obtained within 30 min after adding the Stop Solution using Multiskan FC 3.1 microplate reader (ThermoFisher Scientific, Waltham, MA, USA). Values are reported in ng/mL of ApoA1 calculated using a standard curve produced with dilutjons of purified ApoA1 provided with the kit. The standard curve is presented in Appendix A.

## 3. Results

### 3.1. Cholesterol Recognition/Interaction Amino Acid Consensus (CRAC) Motifs in SIDT2

We have previously identified two CRAC motifs on the sequence of SIDT1 and SIDT2 [3,4]. These CRAC motifs are highly conserved among members of the SIDT family from different species. Figure 1 illustrates the predicted secondary structure of SIDT2, showing its 10 transmembrane domains and the 2 CRAC motifs, one in the long amino terminal region and the second one in the 7th transmembrane domain (Figure 1A). Figure 1B shows an alignment of the sequences from the second CRAC motif, illustrating the highly conserved amino acids V172, Y176, R179 and R/K182 from the first CRAC motif and L571, Y576 and R580 from the second CRAC motif. We have previously shown that replacing Y176 with the amino acid alanine prevents SIDT1 from translocating to the plasma membrane upon cholesterol depletion in culture cells. And, replacing both Y176 (first CRAC motif) and Y576 (second CRAC motif) eliminates cholesterol binding to SIDT1 [4]. The single amino acid mutation in the second CRAC motif (CRAC-2) used in this study is illustrated in Figure 1C, corresponding with a replacement of Y650 for A in SIDT2.

CRAC motifs are constituted by a short linear sequence fulfilling the following criteria (N-terminus to C-terminus direction): a branched apolar L or V residue, followed by a segment containing 1–5 of any residues, then an aromatic residue that is mandatory Y, then again a segment containing 1–5 of any residues, and finally a basic K or R (L/V)-X1−5-(Y)-X1−5-(K/R) [20]. Most importantly, disrupting the Y residue, and even replacing it with another aromatic residue, prevents cholesterol-CRAC motif interactions [21]. This is the main reason why we decided to replace the Y176 and Y576 with A in our previous studies and showed that cholesterol association to SIDT1/2 is indeed abolished by this single amino acid replacement [3,4].

In a recent genome-wide association study, we identified a SIDT2 functional variant present in the Mexican population which was associated with HDL-C levels and premature coronary artery disease (CAD) [8]. This variant was also associated with increased levels of ApoA1 in the serum [8].

ApoA1 is the major component of HDL-C and participates in the production of high-density lipoprotein (HDL) particles [9]. HDL-C plays a critical role in the reverse cholesterol transport (RCT), a process that brings cholesterol from peripheric tissues into the liver [10].

### 3.2. SIDT2 and ApoA1 Associate and Translocate to the Plasma Membrane upon Cholesterol Removal

Since the GWAS studies showed that a SIDT2 variant was associated with high levels of ApoA1, in the present study we decide to investigate possible interactions between both proteins (SIDT2 and ApoA1). For this purpose, we expressed SIDT2 fused to the green fluorescent protein (SIDT2-GFP) and ApoA1 fused to the red fluorescent protein OFPSpark (ApoA1-OFPSpark) in HEK293 cells. These two fluorescent proteins are good pairs for Förster resonance energy transfer (FRET) studies.

We found a good FRET efficiency between SIDT2-GFP and ApoA1-OFPSpark (Figure 2A,B). As a control, we expressed in a different set of cells SIDT2-GFP and OFPSpark, and found poor FRET efficiency (Figure 2B). These results indicate that SIDT2-GFP and ApoA1-OFPSpark are within 10 nm of each other, indicating a possible protein–protein association. To confirm this possibility, we conducted co-immunoprecipitation assays and found that both proteins indeed co-immunoprecipitated (Figure 2C). These results strongly suggest an association of SIDT2 and ApoA1.

In previous studies, we have shown that cholesterol removal in living cells by the addition of methyl-b-cyclodextrin (MbCD), an agent that removes cholesterol from the plasma membrane (PM), results in the translocation of SIDT1/2 to the plasma membrane [4]. SIDT2 translocation to the PM is evident in confocal studies by colocalizing SIDT2-GFP with the PM fluorescent dye FM4-64 (Figure 3A,B). The Pearson co-localization coefficient increased from 0.15 (in the absence of MbCD) to 0.75 (45 min after treatment with 5 mM MbCD).

To further confirm the translocation of SIDT2 to the PM, we conducted total internal reflection fluorescence microscopy (TIRFM) experiments with SIDT2-GFP and FM4-64 (Figure 3C). TIRFM allows the imaging of a narrow optical section of less than 100 nm near the bottom of the cell. The PM translocation of SIDT2-GFP after MbCD treatment was also confirmed with biotinylation studies (Figure 3D). Most interestingly, ApoA1-OFPSpark also translocated to the PM after MbCD treatment (Figure 3E), and biotinylation studies confirmed the translocation (Figure 3F).

### 3.3. The Second CRAC Motif in SIDT2 Modulates the Formation of the SIDT2-ApoA1 Complex

To characterize further the time frame of SIDT2-ApoA1 complex translocation to the PM, we conducted time lapse confocal studies in HepG2 human hepatocarcinoma cells expressing SIDT2-GFP and ApoA1-OFPSpark. First, as a control we conducted time lapse confocal studies with HepG2 cells transfected with SIDT2-GFP and ApoA1-OFPSpark in the absence of MbCD (Figure 4A, upper panel and Appendix A). Line plot analysis showed that most of the protein complex is found at the cytosol (Figure 4B, upper panel). Then a different set of cells was exposed to 5 mM MbCD for 45 min. The PM translocation of the SIDT2-ApoA1 complex was evident 20 min after MbCD application (Figure 4A, middle panel and Appendix A). This observation was confirmed by the line plot analysis (Figure 4B, middle panel).

Because disrupting the first CRAC domain from SIDT2 prevents protein translocation to the PM upon MbCD treatment [4], we decided to explore the role of the second CRAC domain in the association of SIDT2-ApoA1 complex and translocation to the PM. When expressing the CRAC mutant SIDT2-mut-GFP and ApoA1-OFPSpark in HepG2 cells, we observed that MbCD application did not prevent the translocation of SIDT2-mut-GFP to the PM as we have previously reported, since this second CRAC motif does not play a role in protein translocation [4]. However, the translocation of ApoA1 to the PM was prevented when expressing the mutant SIDT2-mut-GFP (Figure 4A, lower panel and Appendix A). This was more evident when conducting line plot analysis, since SIDT2-mut translocated to the PM but ApoA1 remained in the cytosol (Figure 4B, lower panel). To further confirm that the association between SIDT2 and ApoA1 was disrupted by mutating the second CRAC domain from SIDT2, we conducted co-immunoprecipitation studies. Indeed, co-immunoprecipitation of ApoA1 was lost when expressing the mutant SIDT2-mut-GFP (Figure 4C).

### 3.4. SIDT2 Enhances the Secretion of ApoA1 by HepG2 Cells

To evaluate the role of SIDT2 wild type and the SIDT2-mut on the secretion of ApoA1 to the extracellular medium, we conducted enzyme-linked immunosorbent assays (ELISA) to measure the amount of endogenous ApoA1 in the medium of HepG2 cells (Materials and Methods).

First, we determined the amount of nonspecific binding of the antibodies to the ELISA plates by conducting experiments in the absence of cells (Figure 4D first bar). This value was considered the basal nonspecific background level (0.67 ± 0.21 ng/mL, *n* = 5, Figure 4D). Since HepG2 cells secrete endogenous ApoA1 [17], we measured the basal ApoA1 secretion by these cells by collecting media from untransfected HepG2 cells (24.85 ± 3.54 ng/mL, *n* = 5, Figure 4D). The overexpression of SIDT2 increased the ApoA1 secreted to 44.23 ± 7.76 ng/mL (*n* = 5).

To evaluate the role of SIDT2 in the secretion of endogenous ApoA1 by human hepatocytes, we conducted RNA interference (RNAi) experiments directed to knockdown SIDT2 (Materials and Methods). The transfection of a specific small interference RNA for SIDT2 (siSIDT2) resulted in a significant reduction in total SIDT2 protein as evaluated by Western blot (Appendix A). This reduction in SIDT2 was not observed when using a scrambled siRNA (siScrambled, Appendix A). The siScrambled or siSDIT2 transfections did not affect the total ApoA1 protein (Appendix A). Knocking down SIDT2 significantly reduced the amount of ApoA1 co-immunoprecipitated with SIDT2 (Appendix A) and the amount of endogenous ApoA1 secreted by HepG2 cells to a value of 9.33 ± 4.32 ng/mL (*n* = 5, Figure 4D). This value represents 37% of the basal ApoA1 secretion (24.85 ± 3.54 ng/mL), strongly suggesting that over 60% of the ApoA1 secretion is independent from SIDT2.

Most interestingly, the overexpression of SIDT2-mut resulted in ApoA1 secretion values indistinguishable from the basal secretion (21.08 ± 4.94 ng/mL, *n* = 5, Figure 4D), indicating that the mutant does not alter ApoA1 secretion by human hepatocytes. This result supports the hypothesis of a possible role for the second CRAC motif in facilitating the formation of the SIDT2-ApoA1 complex and aiding in the secretion of ApoA1.

Figure 5 summarizes the results from the present study, illustrating the SIDT2-dependent and -independent secretion of ApoA1. The transport towards the PM of vesicles containing SIDT2-ApoA1 complex is enhanced when cholesterol is reduced at the PM. Vesicles are fused with the PM and SIDT2 remains inserted into the PM while ApoA1 is secreted to the extracellular media. ApoA1 associates to SIDT2 via the CRAC-2 motif from SIDT2.

## 4. Discussion

SIDT2 is a multifunctional protein with a wide variety of physiological roles, including lipid hydrolytic activity (SIDT2 hydrolyzes ceramides into sphingosine and fatty acids) [5], uptake of RNA and DNA and degradation by the lysosome [6], apoptosis and autophagy [7].

The first CRAC motif present in a presumably extracellular SIDT domain is essential for protein translocation to the plasma membrane (PM) upon cholesterol depletion from the PM [4].

Several studies, including ours, have shown that SIDT proteins reside in intracellular compartments and very little protein is found at the PM [4,5,6]. However, our previous studies have illustrated how SIDT1 moves to the PM upon cholesterol depletion with MbCD treatment. For this translocation, a functional CRAC-1 motif is required [4].

Apolipoprotein A1 (ApoA1) is the major component of HDL and participates in the production of high-density lipoprotein (HDL) particles [9]. HDL-C plays a critical role in the reverse cholesterol transport (RCT), a process that brings cholesterol from peripheric tissues into the liver [10]. In the present study we have shown that SIDT2 and ApoA1 associate forming a protein complex in intracellular compartments. The positive FRET signal between SIDT2 and ApoA1 strongly suggest a direct interaction between the two proteins. This interaction allows the simultaneous translocation of SIDT2 and ApoA1 to the PM upon MbCD treatment. Changing a single amino acid (Y650A) in the CRAC-2 motif dissociates the SIDT2-ApoA1 protein complex and prevents the simultaneous translocation of the complex to the PM (without affecting SIDT2 translocation). This result indicates that the second CRAC motif in SIDT2 (CRAC-2) is required for the association with ApoA1 and the assembly of the complex. The mechanism facilitating the formation of the SIDT2-ApoA1 protein complex remains to be identified, but clearly cholesterol appears to be the partner mediating this association since both proteins (SIDT2 and ApoA1) bind cholesterol.

In a recent genome-wide association study, we identified a SIDT2 functional variant present in the Mexican population associated with high HDL-C levels and reduced premature coronary artery disease (CAD) [8]. This variant was also associated with increased levels of ApoA1 in the serum of individuals expressing this SIDT2 variant [8].

Most interestingly, the SIDT2 variant shows a single amino acid replacement, consisting of a change in I636V [8]. This amino acid is only five residues away from the CRAC-2 motif in SIDT2 (CRAC-2 motif encompasses residues 641–659, Figure 1). Expression of the SIDT2 I636V variant in HEK293 cells results in enhanced cholesterol uptake, indicating a functional increment in cholesterol transport in this SIDT2 variant [8].

In the present study, we have shown that the over expression of SIDT2 in HepG2 cells results in greater secretion of endogenous ApoA1 to the extracellular medium. Most interestingly, this effect was not observed when overexpressing the SIDT2-mut, further pointing to the idea that SIDT2 facilitates the secretion of ApoA1 and that the second CRAC motif (CRAC-2) in SIDT2 is essential for this function. Knocking down SIDT2 reduces the ApoA1 co-immunoprecipitated with SIDT2 without affecting total ApoA1 protein content. SIDT2 protein reduction with the siSDIT2 results in reduced endogenous ApoA1 secretion by human hepatocytes, further supporting the hypothesis that SIDT2 facilitates ApoA1 secretion.

The most favorable hypothesis is that the CRAC-2 motif participates in the SIDT2-ApoA1 complex formation, allowing the transport of ApoA1 to the PM upon cholesterol depletion. This is consistent with our observation that the SIDT2-mut translocate normally to the PM upon MbCD treatment, but ApoA1 does not (Figure 4 and Appendix A). Since SIDT2 has 8–11 predicted transmembrane regions [5], it must be at all times inserted into a cell membrane. Thus, it is feasible that ApoA1 (either lipidated or in its lipid-free form, or both) associates to the vesicles carrying SIDT2 and travels in this form towards the PM.

The exact mechanism involved in the secretion of ApoA1 by hepatocytes is poorly understood, but a fraction of secreted ApoA1 is in a lipidated state, while the remaining ApoA1 is secreted in a lipid-free form [17,18]. We have not established yet which of the ApoA1 forms (lipidated, lipid-free or both) is being secreted in association with SIDT2. Further studies analyzing the secreted forms of ApoA1 may help to elucidate the molecular mechanism responsible for the SIDT2-ApoA1 complex translocation and facilitation of ApoA1 secretion to the extracellular medium. What is noteworthy is that a substantial fraction of ApoA1 appears to be secreted independently from SIDT2 (Figure 4D). Whether this fraction is in the lipidated or lipid-free form remains to be established.

## 5. Conclusions

The SIDT family of transporters is present in species from nematodes to humans. *C. elegans* carries a single gene encoding SIDT (ZK721.1/tag-130) [3], while in mammals there are two genes, SIDT1 and SIDT2. SIDT2 is a multifunctional protein with a wide variety of physiological roles, including lipid hydrolytic activity (SIDT2 hydrolyzes ceramide into sphingosine and fatty acid) [5], uptake of RNA and DNA and degradation by the lysosome [6], apoptosis and autophagy [7].

In the present study, we have provided evidence indicating that SIDT2 forms a protein complex with ApoA1. The second CRAC motif in SIDT2 (CRAC-2) is required for the complex formation. Upon cholesterol depletion from the plasma membrane (PM), the SIDT2-ApoA1 complex travels to the plasma membrane, resulting in the enhanced secretion of ApoA1. The SIDT2-mediated secretion of ApoA1 represents about 40% of the total ApoA1 secretion in hepatocytes. This study uncovers a newly discovered role for SIDT2 in aiding the secretion of ApoA1 by human hepatocytes.

## Figures and Tables

**Figure 1 cells-12-02353-f001:**
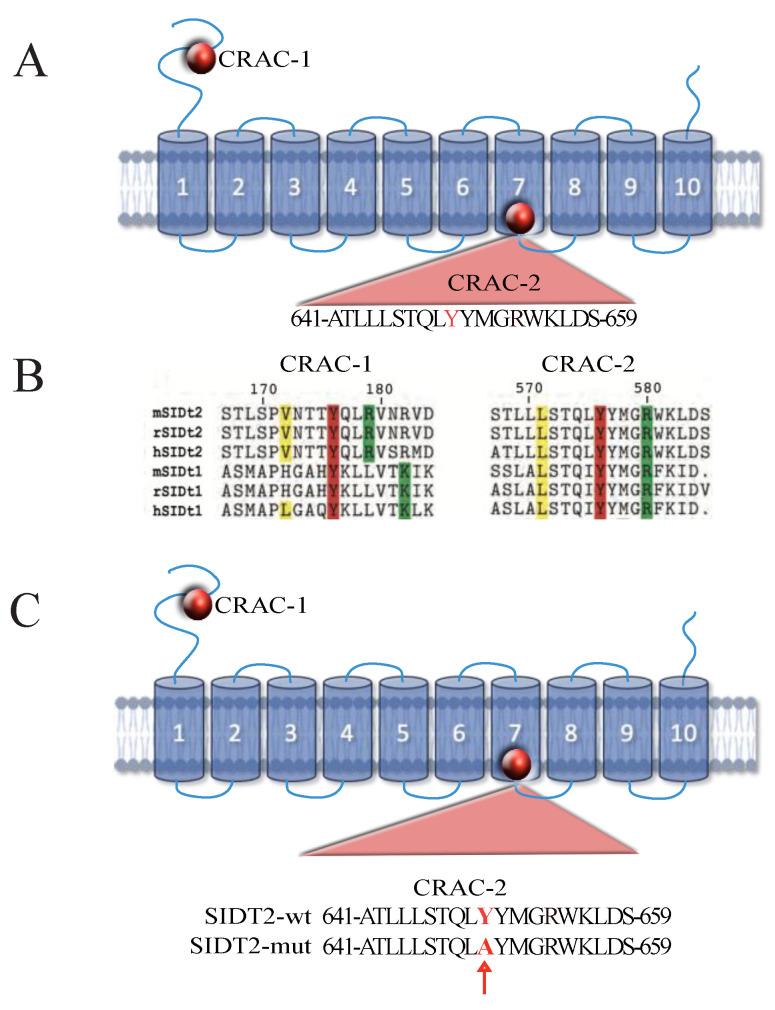
CRAC motifs in SIDT family. (**A**)The putative topology of SIDT2. SIDT2 is predicted to have between 8 and 11 transmembrane domains [5]. For illustration purposes, 10 are displayed. Red circles indicate the relative position of both CRAC motifs. Insert shows the sequence for the transmembrane CRAC motif (CRAC-2) in SIDT2. (**B**) Alignment of the conserved extracellular and transmembrane CRAC motifs (L/V-X(1–5)-Y-X(1–5)-R/K) in SIDT homologue proteins. Numbers indicate amino acid position in Chup-1 from *C. elegans* [3]. Colored amino acid residues indicate residues present in all CRAC motifs. Color code show positively charged (green), aliphatic (yellow) and aromatic (red) residues. The tyrosine is essential for cholesterol binding. m: Mus musculus; r: Rattus norvegicus; h: Homo sapiens. (**C**) The amino acid replacement to mutate the second CRAC motif (CRAC-2) in SIDT2. Red arrow points to the single amino acid replacement of tyrosine (Y) for alanine (A).

**Figure 2 cells-12-02353-f002:**
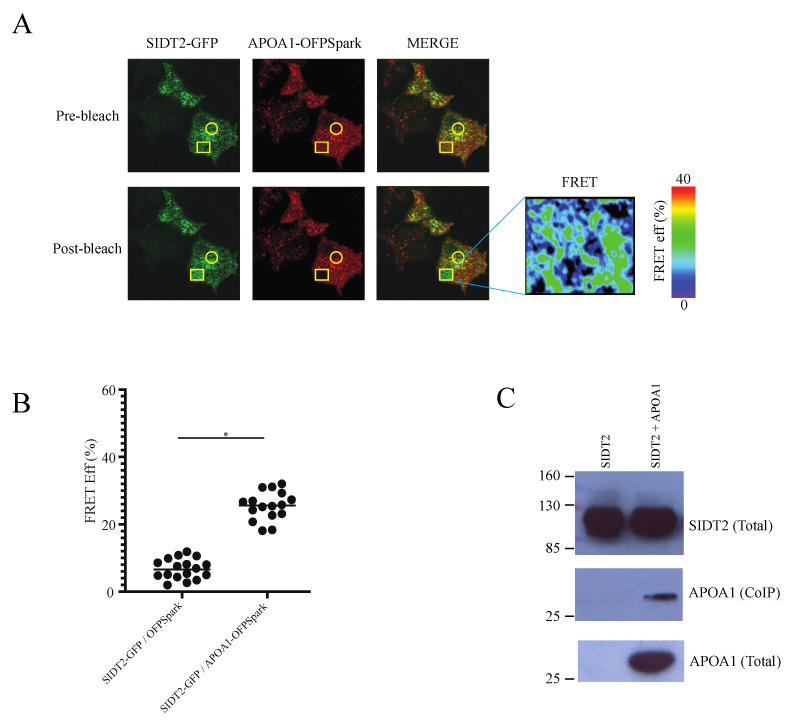
SIDT2 and ApoA1 associate in a protein complex. (**A**) Confocal microscopy Förster resonance energy transfer (FRET) studies in HEK293 cells expressing SIDT2-GFP (green) and ApoA1-OFPSpark (red). Yellow circle indicates the area that was not photobleached (baseline) and the yellow square indicates the area where acceptor photobleaching was conducted to calculate FRET efficiency. The insert to the right shows the rectangular area with color mapped FRET efficiency. (**B**) FRET efficiency calculated from multiple cells (each dot is a cell). Horizontal lines show the mean efficiency for each group. The control group was cells expressing SIDT2-GFP and soluble OFPSpark to determine the amount of nonspecific FRET resulting from the possible association of fluorescent proteins. Asterisks indicate *p* < 0.02. (**C**) Coimmunoprecipitation studies conducted with HEK293 cell extracts with cells expressing SIDT2-GFP and ApoA1. SIDT2-GFP was immunoprecipitated using a specific anti-GFP antibody. SIDT2 and ApoA1 identified in Western blot using selective antibodies (Materials and Methods).

**Figure 3 cells-12-02353-f003:**
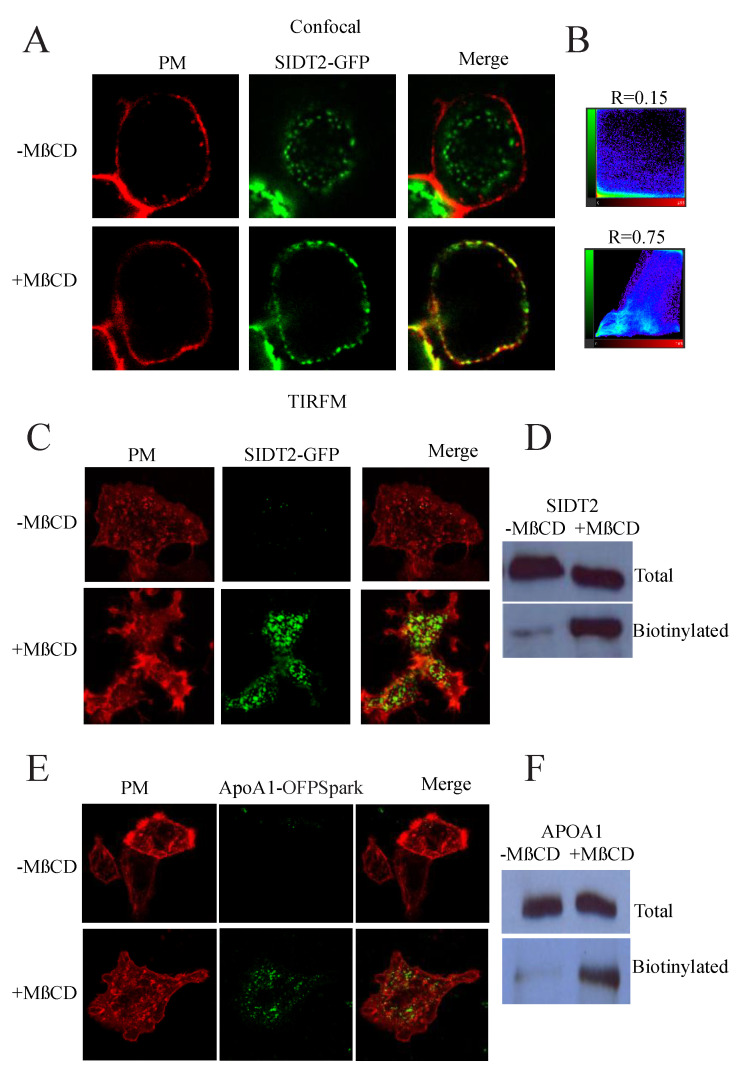
SIDT2 and ApoA1 translocate to the plasma membrane upon cholesterol removal. (**A**) Confocal microscopy studies using HEK293 cells expressing SIDT2-GFP with the plasma membrane (PM) labeled with FM4-64 in the absence and presence of MbCD (5 mM for 45 min). Yellow indicates colocalization between SIDT2-GFP (green) and FM4-64 (red). (**B**) Colocalization studies using Pearson’s colocalization coefficient analysis. (**C**) total internal reflection fluorescence microscopy studies in HEK293 cells expressing SIDT2-GFP (green) with the PM labeled with FM4-64 (red) in the absence and presence of MbCD (5 mM for 45 min). (**D**) Biotinylation studies conducted in intact cells to assess the fraction of SIDT2-GFP exposed to the extracellular compartment. Blots show total and biotinylated SIDT2 fractions in the absence and presence of MbCD (5 mM for 45 min). (**E**) Total internal reflection fluorescence microscopy studies in HEK293 cells expressing ApoA1-OFPSpark (green) with the PM labeled with FM4-64 (red) in the absence and presence of MbCD (5 mM for 45 min). (**F**) Biotinylation studies conducted in intact cells to assess the fraction of ApoA1 exposed to the extracellular compartment (full length blots shown in Appendix A). Blots show total and biotinylated ApoA1 fractions in the absence and presence of MbCD (5 mM for 45 min).

**Figure 4 cells-12-02353-f004:**
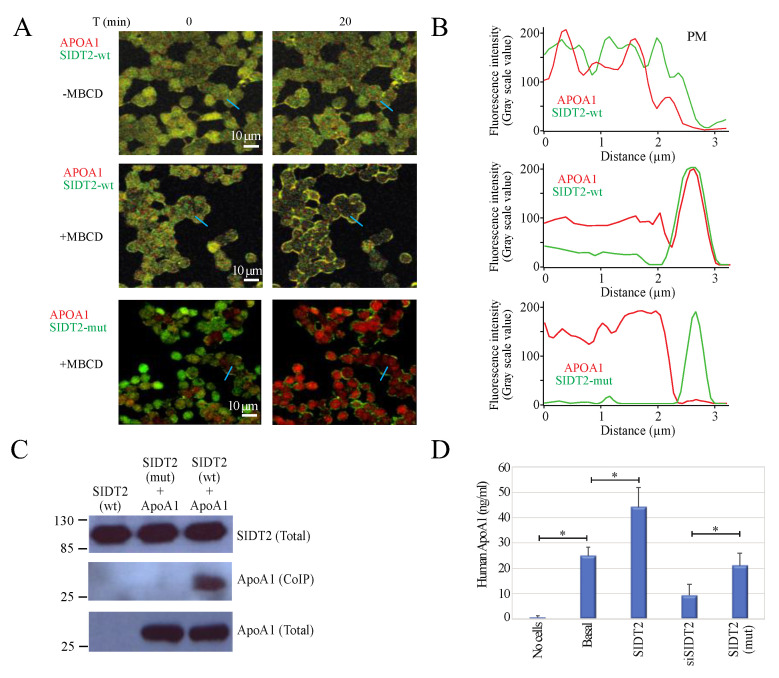
Mutation of the second CRAC motif in SIDT2 disrupts SIDT1-ApoA1 complex formation and translocation. (**A**) Confocal microscopy time lapses conducted in HepG2 cells expressing SIDT2-GFP (green) and ApoA1-OFPSpark (red) in non-stimulated conditions (absence of MbCD, upper panels) and HepG2 cells expressing SIDT2-GFP (green) and ApoA1-OFPSpark (red) at 0 and 20 min after continuous exposure to 5 mM MbCD (middle panels). The lower panels show confocal microscopy images conducted in HepG2 cells expressing SIDT2-mut-GFP (green) and ApoA1-OFPSpark (red). SIDT2-mut-GFP is the mutant with the CRAC-2 motif disrupted. For full time lapses, please refer to Appendix A. (**B**) Line plot analysis for the cells illustrated in panel A (marked with a blue line). (**C**) Coimmunoprecipitation studies conducted with HEK293 cells expressing only SIDT2-GFP or SIDT2-GFP and ApoA1 or SIDT2-mut-GFP and ApoA1 (full-length blots shown in Appendix A). (**D**) Enzyme-linked immunosorbent assays (ELISA) with HepG2 cells under the following conditions: with no cells in the assay (no cells), with HepG2 untransfected cells (basal), with HepG2 overexpressing ApoA1, with HepG2 overexpressing SIDT2 and ApoA1 and finally with HepG2 overexpressing SIDT2-mut and ApoA1. Values are the mean ± standard deviation obtained from 4 replicas repeated in 5 independent experiments (20 total values for each condition). Asterisks indicate *p* < 0.02.

**Figure 5 cells-12-02353-f005:**
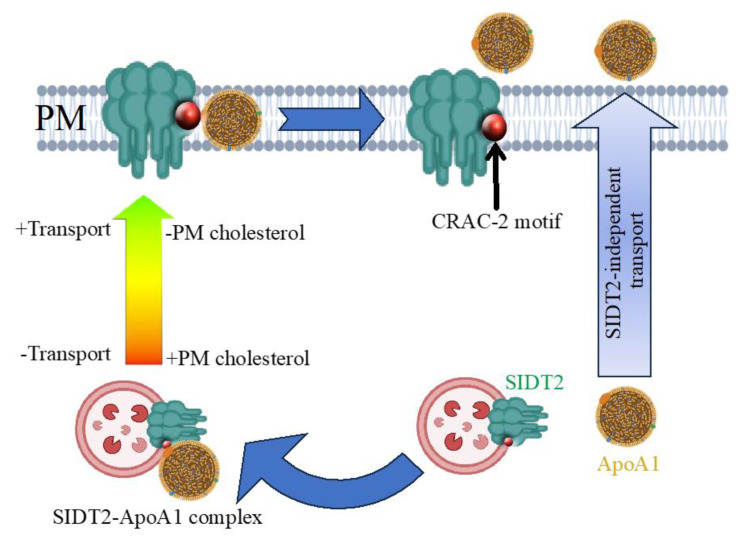
SIDT2-dependent and independent secretion of ApoA1. The two mechanisms of ApoA1 secretion, one dependent on SIDT2 association (left side) and one SIDT2-independent (right side). SIDT2 is depicted on a vesicle, which may be a lysosome or another intracellular vesicle. The SIDT2-dependent ApoA1 secretion is enhanced when plasma membrane (PM) cholesterol is reduced. Upon cholesterol reduction at the PM, SIDT2 is inserted into the PM and ApoA1 is secreted to the extracellular medium. The red circle in the SIDT2 image represents the CRAC-2 motif, which plays a role in the assembly of the SIDT2-ApoA1 complex.

## Data Availability

Data available upon request.

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
