# Peer review of "SIDT2 Associates with Apolipoprotein A1 (ApoA1) and Facilitates ApoA1 Secretion in Hepatocytes"

_cells, 2023, doi:10.3390/cells12192353_

Round 1
Reviewer 1 Report
SIDT2 is a lysosomal membrane protein involved in cholesterol transfer between membranes. GWAS studies linked SIDT2 with HDL-cholesterol levels. SIDT2 contains two potential cholesterol-binding domains called CRAC motif. Using overexpression systems in the human hepatoma cell line HepG2, this work showed that SIDT2 forms a protein complex with apoA1. The authors then used a variety of cell imaging and biochemical methods and showed that the complex formation between SIDT2 and apoA1 requires one of the two CRAC motifs present within SID2.
Currently, how ApoA1 within the cells is secreted from hepatocytes is largely unknown. The results provided here are suggestive because only the overexpression system is used ( i.e., no information on liver cells lacking endogenous SIDT2 is available), and the cells must be treated with cyclodextrin. These deficiencies should be pointed out in the Abstract and in the Discussion, but the results are novel and interesting. This manuscript is worthwhile publishing.
I also have one question- SID2 is known to contain lipid hydrolytic activity. Did the authors test if SID2 that contains the CRAC domain mutation expresses normal or altered lipid hydrolytic activity?
Other comments-
Please report the source of ApoA1-OFPSpark
Author Response
REVIEWER 1
Currently, how ApoA1 within the cells is secreted from hepatocytes is largely unknown. The results provided here are suggestive because only the overexpression system is used ( i.e., no information on liver cells lacking endogenous SIDT2 is available), and the cells must be treated with cyclodextrin. These deficiencies should be pointed out in the Abstract and in the Discussion, but the results are novel and interesting. This manuscript is worthwhile publishing.
AUTHORS: Following your pertinent suggestion we have included a sentence explaining that cholesterol depletion from the PM is required for the transport of the SIDT2-ApoA1 complex to the plasma membrane. We do not see the requirement of cyclodextrin as a deficiency, but rather as a mechanism to stimulate the transport of SIDT2 to the PM.
I also have one question- SID2 is known to contain lipid hydrolytic activity. Did the authors test if SID2 that contains the CRAC domain mutation expresses normal or altered lipid hydrolytic activity?
AUTHORS: We did not test the lipid hydrolytic activity of SIDT2. Although your suggestion about testing the catalytic activity of SIDT2 mutant is very compelling, we think this deviate from the main intention of this publication, which is to explore the complex formed by SIDT2-ApoA1. We would like to explore this possibility with detail, which requires a dedicated report and different experiments.
Other comments-
Please report the source of ApoA1-OFPSpark
AUTHORS: We have included a paragraph in the methods section indicating the source for ApoA1-OFPSpark (Sino Biological Inc.).
Reviewer 2 Report
Please see attached WORD document.

Suggested grammatical edits are included in the review.
Author Response
REVIEWER 2
This study is a follow-up study to earlier work done by this group that identified cholesterol recognition (CRAC) motifs in the lysosomal transmembrane proteins SIDT1 and SIDT2, which act to promote transport of cholesterol between membranes. A GWAS study determined that a mutation in close proximity to the CRAC-2 motif in SIDT2 is associated with elevated plasma HDL-C and apoAI. In this study, the authors investigate the interaction of SIDT2 and SIDT2-mut in which the transmembrane CRAC-2 domain is mutated (Y650A) with apoAI (Fig 1 model). They show co-localization of apoAI and SIDT2 in lysosomes (Fig 2) when these are both expressed in HEK293 cells transfected with both proteins, and movement of both proteins to the plasma membrane when cells are depleted of cholesterol upon treatment with MbCD (Fig 3). More physiologically relevant studies are with the HepG2 cells, which have endogenous apoAI. Fig 4 A, B and the supplemental videos provide compelling evidence for the role of the CRAC-2 domain in translocation of apoAI to the plasma membrane upon cholesterol depletion, and Fig 4D demonstrates that increased WT-SIDT2 can promote apoAI secretion from hepatocytes transfected with excess apoAI, thus linking this work to the GWAS study. However, the authors did not study the effect of SIDT2 on endogenous apoAI secretion, nor did the study the GWAS variant, so the latter remains for future work.
AUTHORS: Following your pertinent suggestion, we have included experiments showing the effect of SDIT2 knockdown in hepatocytes on the secretion of endogenous ApoA1 (New figure 4D and supplemental figure 3). Indeed, the GWAS variant will be explored in future experiments.
The GWAS study gives credence for physiological relevance of SIDT2 in HDL metabolism and CAD. However, since apoAI is normally synthesized and processed through the ER and Golgi, and secreted from vesicles, it is not considered to be a PM protein. The authors should add to their discussion their model for how apoAI, bound to a transmembrane domain of SIDT2, could play a role in cholesterol homeostasis. Overall this paper presents intriguing, novel results and is worthy of publication.
AUTHORS: We do not think that ApoA1 is a membrane resident protein. Our results suggest that SIDT2 facilitates the transport of ApoA1 (either in lipidated or lipid-free forms) towards the PM and facilitates secretion of ApoA1. SIDT2 remains at the PM and it is not secreted with ApoA1. Following your pertinent suggestion, we have included a model explaining the role of the SIDT2-ApoA1 complex in the transport and secretion of ApoA1 (new figure 5).
However, I suggest the following edits/revisions:
Title
Line 2: replace “to” with ‘with’ to read associate with apoliprotein A1
AUTHORS: Changed.
Introduction
Line 60: replace ‘to’ with ‘with’
AUTHORS: Changed.
Line 67: to ref 9 and 11, add PMID 34800735, Sean Davidson’s HDL proteome website reference.
AUTHORS: Added.
Line 69: While HDL-C is inversely related to risk for ASCVD, more recent epidemiological data shows that very high HDL-C is associated with increased all-cause mortality and CVD. A good reference for this is PMID 29326314, Hamer, et al. 2018.
AUTHORS: Added.
Line 76: The John Parks lab was the first to show that some apoAI is secreted in lipid-free form, and this reference should be added to ref 15: PMID 11792720, Parks, 2002.
AUTHORS: Reference added. Now is reference number 17 in the revised version.
Line 84: add’ provide’ and ‘of’ to read ‘results provide evidence of a ….”
AUTHORS: Changed. Now is in line 91 of the revised version.
Line 85: delete s from suggests
AUTHORS: Deleted. Now in line 92 of the revised version.
METHODS
Line 98: Please add to the Methods how you created the SIDT2-mut that was used in this study.
AUTHORS: A detailed description of how we created the mutant is in our previous publications. We have added references in the methods section.
Line 99: replace Apoa1 with ApoA1, i.e. capital A, twice. This needs to be done here and in several other places in the Methods, i.e. line 115, 118 and 132.
AUTHORS: We revised the entire manuscript and corrected all to capital A.
Line 113: enclose ThermoFisher in the parentheses: (ThermoFisher, Waltham, MA).
AUTHORS: Enclosed.
Line 115: Saint, not San
AUTHORS: Changed to Saint.
Line 117: add d to prepare to be prepared
AUTHORS: Added.
Line 122: again, Saint, not San
AUTHORS: Changed to Saint.
Line 158: Does SIDT2 promote secretion of endogenous apoAI? The authors should add to their study HepG2 cells transfected with SIDT2 (without apoAI transfection) to answer this question. If so, then also they should determine the effect of the SIDT2-mut (without apoAI transfection) to show the effect of CRAC-2 mutant on endogenous apoAI secretion.
AUTHORS: This is an excellent suggestion. We have included the data in new figure 4D and new supplemental figure 3.
Line 164, regarding the ELISA assay for measurement of apoAI in the cell medium: Please provide details on how long the media was incubated in the 96-well plates prior to addition of antibodies? Do the authors assume that the apoAI bound to the Opti-MEM ECM that was used to coat the wells? As supplemental information, the authors should provide supporting documentation of the ELISA protocol, with a Standard Curve, detection limit and range.
AUTHORS: We have included details on the incubation times in the revised methods section. We have included the standard curve using purified ApoA1 (new supplemental figure 3). Furthermore, we have changed the data reported in figure 4D to use ApoA1 concentration based on the standard curve.
RESULTS
The first few paragraphs of the Results section discussed previous/published work and should be moved and combined with the Introdution
AUTHORS: Combined.
Lines 174 to 216: moved these paragraphs to the Introduction, but avoid repetition.
AUTHORS: We have rewritten the results section to accommodate your suggestion.
Line 199: In the Figure 1 legend, add a description of section C. This is missing.
AUTHORS: Added.
Line 219: Insert to read as follows: “Since the GWAS studies showed that a SIDT2 variant…
AUTHORS: Changed.
Line 220: edit to : study we decide to explore any possible investigated possible interactions between both proteins (SIDT2 and apoAI).
AUTHORS: Changed.
Line 224: replace ‘both’ with ‘These two’
AUTHORS: Replaced.
Line 229: You show that anti SIDT2 co-IPs apoAI. Does anti apoAI co-IP SIDT2? For this work you used HEK293 cells transfected with both SIDT2 and apoAI. Please also add co-IP data from the HepG2 cells, i.e. a more physiologically relevant cell time. In the HepG2 cells, does SIDT2 co-IP endogenous apoAI, or does this association happen only in the presence of excess transfected apoAI?
AUTHORS: This is a very good suggestion. We have included CoIP studies with HepG2 cells (new supplemental figure 3).
Line 231: delete d to be an rather than and
AUTHORS: Deleted.
Line 233: from, not form
AUTHORS: Changed.
Line 246, Figure 2: add scale bars to Fig 2A. Note in the legend of the text that previous work showed that SIDT2 localizes to lysosomes, so that the vesicles in Fig 2A are presumably lysosomes.
AUTHORS: Scale bars added to all figures. Some of the vesicles shown in Figure 2A are most likely lysosomes, but SIDT2 is present in other intracellular organelles (PLoS One. 2012;7(3):e33962. doi: 10.1371/journal.pone.0033962. Epub 2012 Mar 27.PMID: 22479487).
Line 254: C should be bold.
AUTHORS: Changed to bold.
Line 255: HEK293, not HEL293.
AUTHORS: Changed to HEK293 cells.
Line 257: The last sentence should be moved to the legend for section B. Add s to read indicates.
AUTHORS: Moved. S added.
Line 261: Very nice data to show colocalization to the PM. Please add scale bars to the images.
AUTHORS: Thank you.
Line 264: Add s to indicates
AUTHORS: Added.
Line 281: The supplemental videos are very nice.
AUTHORS: Thank you.
Line 302: Beautiful study with the HepG2 cells, Figure 4 A, B and D.
AUTHORS: Thank you.
Line 308: Please give the size of the scale bars in Figure 4A.
AUTHORS: Scales added.
Line 312: Fig 4D, on the effect of SIDT2 on apoAI secretion was only done in the presence of excess apoAI, i.e. in HepG2 cells transfected to overexpress apoAI. The authors should also look at SIDT2 effect of secretion of endogenous apoAI. This is a repeat of the comment from above:
Line 158: Does SIDT2 promote secretion of endogenous apoAI? The authors should add to their study HepG2 cells transfected with SIDT2 (without apoAI transfection) to answer this question. If so, then also do the experiments with the SIDT2-mut (without apoAI transfection) to show effect of the mutant on endogenous apoAI secretion.
AUTHORS: This is an excellent suggestion. We have conducted RNAi experiments to knock down endogenous SIDT2 in hepatocytes and show that ApoA1 CoIP is significantly reduced (new supplemental figure 3).
Line 352: apoAI is the major protein of HDL particles, so rewrite the first sentence to read: “Apolipoprotein A1 (ApoAI) is the major protein of HDL and participates….” Note that the particle is HDL, and cholesterol is one of its components, so in this case use HDL, not HDL-C.
AUTHORS: Rewritten in the revised manuscript.
Line 357: replace ‘both’ with ‘the two’
AUTHORS: Replaced.
Line 362: replace ‘to’ with ‘with’
AUTHORS: Replaced.
Line 372: The GWAS mutant is of very much clinical interest. Is it known if this variant has altered cholesterol or apoAI association? Do the authors plan future experiments to study this variant?
AUTHORS: In our previous study we showed that SIDT2 variant promotes increased cholesterol uptake in HEK293 cells when compared with the wild type SIDT2. We are currently working on new experiments to elucidate the effect of the SIDT2 variant in the formation of the SIDT2-ApoA1 complex and in the secretion of ApoA1.
Line 385: add ‘it’ after [5], to read ‘it must be’
AUTHORS: Added.
Line 387: add s to travels……..
AUTHORS: Added.
Line 388: Add to this paragraph a model of the mechanism by which SIDT2 promotes apoAI secretion. The apoAI that binds SIDT2 is presumably apoAI that has been taken up by cells via endocytosis, since this is the lysosomal compartment. So do the authors think this a salvage pathway to reutilize apoAI in formation of nascent HDL? What portion of total cell apoAI is found in association with SIDT2?
AUTHORS: Following your pertinent suggestion we have added a new model (figure 5) to explain how the SIDT2-ApoA1 complex may facilitate ApoA1 secretion in the hepatocyte.
Line 390: to reference 15, add the Parks reference noted above in the Introduction: PMID 11792720, Parks, 2002
AUTHORS: Added.
Line 391: rewrite this line to read ‘is being secreted in association with SIDT2’, deleting ‘the collaboration of’
AUTHORS: Rewritten in the revised manuscript.
Round 2
Author Response
We have modified the discussion to merged some of the text with the introduction, as suggested by the referee. We have rewritten the discussion section to avoid repetition. We included a conclusion section highlighting the main findings from our study.
We have included also the no conflict of interest statement.